# COVID-19 pathophysiology may be driven by an imbalance in the renin-angiotensin-aldosterone system

Susanne Rysz [1,2], Jonathan Al-Saadi[3], Anna Sjöström [4,5], Maria Farm [4,5], Francesca Campoccia Jalde[2,4], Michael Plattén [3,6], Helen Eriksson[7], Margareta Klein[4,8], Roberto Vargas-Paris[4,9], Sven Nyrén[4,9], Goran Abdula[4,10], Russell Ouellette [3,6], Tobias Granberg [3,6], Malin Jonsson Fagerlund [2,11] & Johan Lundberg [3,6 ✉]

SARS-CoV-2 uses ACE2, an inhibitor of the Renin-Angiotensin-Aldosterone System (RAAS), for cellular entry. Studies indicate that RAAS imbalance worsens the prognosis in COVID-19. We present a consecutive retrospective COVID-19 cohort with findings of frequent pulmonary thromboembolism (17%), high pulmonary artery pressure (60%) and lung MRI perfusion disturbances. We demonstrate, in swine, that infusing angiotensin II or blocking ACE2 induces increased pulmonary artery pressure, reduces blood oxygenation, increases coagulation, disturbs lung perfusion, induces diffuse alveolar damage, and acute tubular necrosis compared to control animals. We further demonstrate that this imbalanced state can be ameliorated by infusion of an angiotensin receptor blocker and low-molecular-weight heparin. In this work, we show that a pathophysiological state in swine induced by RAAS imbalance shares several features with the clinical COVID-19 presentation. Therefore, we propose that severe COVID-19 could partially be driven by a RAAS imbalance.

[1] Department of Medicine Solna, Karolinska Institutet, Stockholm, Sweden. [2] Function Perioperative Medicine and Intensive Care, Karolinska University Hospital, Stockholm, Sweden. [3] Department of Clinical Neuroscience, Karolinska Institutet, Stockholm, Sweden. [4] Department of Molecular Medicine and Surgery, Karolinska Institutet, Stockholm, Sweden. [5] Department of Clinical Chemistry, Karolinska University Hospital, Stockholm, Sweden. [6] Department of Neuroradiology, Karolinska University Hospital, Stockholm, Sweden. [7] Department of Sociology, Stockholm University Demography Unit, Stockholm University, Stockholm, Sweden. [8] Department of Radiology Huddinge, Karolinska University Hospital, Stockholm, Sweden. [9] Department of Radiology Solna, Karolinska University Hospital, Stockholm, Sweden. [10] Department of Clinical Physiology, Karolinska University Hospital, Stockholm, Sweden. [11] Department of Physiology and Pharmacology, Karolinska Institutet, Stockholm, Sweden. ✉email: j.lundberg@ki.se

Severe acute respiratory syndrome coronavirus (SARS-CoV) and SARS-CoV-2 enter human cells through cell surface angiotensin-converting enzyme 2 (ACE2) receptors[1]. SARS-CoV downregulates ACE2, with a subsequent increase in angiotensin II (ANGII) levels[2], which potentially creates a dysregulation with overactivation of the Renin-Angiotensin-Aldosterone System (RAAS)[3]. The RAAS is a hormonal system contributing to the control of circulating blood volume, blood pressure, and is often pharmaceutically manipulated in hypertensive disease[4]. Current clinical management guidelines for the coronavirus disease 2019 (COVID-19) are mainly centered around the assumption that SARS-CoV-2 directly results in an acute lung parenchymal disease[5,6].

Initially, much of the scientific discussion regarding RAAS and COVID-19 treatment focused on angiotensin-converting enzyme inhibitors (ACEi) or angiotensin receptor blockers (ARB)[7–9], and the hypothesis that ACEi and ARB might increase ACE2 expression and thus increase infection risk and severity[10,11]. Observational data now supports the continued use of ACEi and ARB in COVID-19 patients[12]. Some retrospective cohorts also observed lower all-cause mortality for patients on ACEi and ARB[13]. However, no studies have yet been published concerning the de novo initiation of ARB in COVID-19.

One of the commonly proposed underlying pathophysiological mechanisms of COVID-19 is a viral acute respiratory distress syndrome (ARDS) coupled with high levels of cytokines that subsequently lead to a cytokine-release syndrome[5]. However, although patients with severe COVID-19 can meet the ARDS Berlin definition[14], several reports from our intensive care units (ICU), as well as described by others, have noted relatively well-preserved lung mechanics that is not associated with a typical ARDS-pattern[6,15]. Furthermore, pro-inflammatory cytokines have been found to be lower than the levels expected during cytokine-release syndrome[16,17]. Meanwhile, a preliminary report from Wuhan and a more extensive retrospective cohort described that ANGII levels in COVID-19 were markedly elevated and closely associated with lung injury, while renin levels remained similar to controls[18,19]. This suggests that plasma ANGII elevation may be related to the SARS-CoV-2 pathophysiology. In two other studies of another cohort, normal ANGII but lower circulating ANG(1-7) levels were associated with a more severe outcome in COVID-19[20,21].

Another notable paradox in COVID-19 is that viral RNA is seldomly detected in patient plasma; highlighted by one study that reported viremia in only 6/41 patients and 2/15 ICU patients[22]. Despite the low viral detection in the systemic circulation, currently published histopathological reports have found end-organ failure in the lung, kidney, heart, and small intestines, amongst others[23–28]. We initiated this translational project to investigate the degree to which a disturbance in the RAAS could possibly contribute to the pathophysiological syndrome of COVID-19.

In this work, we report high frequencies of thromboembolism, signs of high pulmonary artery pressure and in one case lung perfusion disturbances in a consecutive retrospective COVID-19 imaging cohort. We further sought to develop a large animal model that demonstrates the systemic effects of RAAS imbalance, which significantly raise pulmonary artery pressures, place the animal in a hypercoagulation state, significantly reduce blood oxygenation and lead to lung perfusion disturbances that share similarities with COVID-19.

## Results

**Retrospective imaging cohort.** We retrospectively analyzed all consecutive computed tomography pulmonary angiography (CTPA) examinations acquired at Karolinska University Hospital in Huddinge, Stockholm, Sweden, in patients with reverse transcription-polymerase chain reaction (RT-PCR) confirmed SARS-CoV-2 infection. In total, 339 CTPA examinations were performed in 289 adult patients (Fig. 1). The mean age of the cohort was $59 \pm 16$ (mean $\pm$ SD) years and 72% were male. Motion artifacts or other technical issues prohibited pulmonary artery (PA) diameter measurements in 47 examinations, resulting in 292 analyzed scans. The mean PA diameter was $29.2 \pm 4.3$ mm (mean $\pm$ SD), with 174 of 292 (60%) scans demonstrating an abnormally wide PA diameter of $\geq 28$ mm, suggestive of pulmonary hypertension[29]. Pulmonary macro-thrombosis/embolism was present in 56 of 336 examinations (17%, three scans were not interpretable due to technical quality issues). We then analyzed a subset of the CTPA cohort that had also undergone echocardiography ($n = 50$). Echocardiography indicated high PA pressure: maximal tricuspid regurgitation velocity $3.0 \pm 0.43$ m/s ($n = 38$; normal reference $\leq 2.8$ m/s); estimated systolic pulmonary artery pressure $47 \pm 14$ mmHg ($n = 38$; normal reference <36 mmHg); right ventricular outflow tract acceleration time $57 \pm 43$ ms ($n = 17$; normal reference >100 ms)[30,31]. Maximal tricuspid regurgitation velocity and estimated systolic pulmonary artery pressure were not measurable in 12 patients due to insufficient tricuspid valve regurgitation Doppler signal.

These findings are corroborated by invasive pressure measurements from a SARS-CoV-2 RT-PCR-positive 58-year-old male ICU patient. A pulmonary artery Swan-Ganz catheter had been placed on ICU day 25 with recorded systolic and diastolic pulmonary artery pressures of $51 \pm 2.9/11 \pm 3.2$ mmHg (mean $\pm$ SD) and a $SvO_2$ of $62.4 \pm 1.3\%$ (mean $\pm$ SD) during the first hour after placement. Upon admission to the ICU, the patient's clinical chemistry values were $PaO_2$ 8.0 kPa (reference range 8.0–13 kPa) and $PaCO_2$ 5.1 kPa (reference range 4.6–6.0), C-reactive protein 66 mg/L (reference range <3 mg/L), interleukin-6 95 ng/L (reference range <7 ng/L), TNF-α 9.8 ng/L (reference range <12 ng/L), AST 0.53 μkat/L (reference range <0.76 μkat/L), and D-dimer 0.26 mg/L FEU (age-adjusted cut-off <0.58 mg/L FEU).

**Clinical chemistry characteristics of imaging cohort.** We retrospectively collected clinical chemistry data in the CTPA cohort based on the blood sampling closest to the exam date, within $\pm$ three days from the first CTPA examination. The median of D-dimer was 1.13 mg/L FEU (lower quartile 0.66, upper quartile 3.0, reference range <0.5 mg/L FEU, $n = 257$), fibrinogen 6.3 g/L (lower quartile 4.6, upper quartile 7.4, reference range 2–4.2 g/L, $n = 91$), interleukin-6 76 ng/L (lower quartile 38, upper quartile 151, reference range <7, $n = 147$), and TNF-α 13 ng/L (lower quartile 10, upper quartile 19, reference range <12 ng/L, $n = 78$).

**MRI lung perfusion.** A clinical MRI lung perfusion scan had been performed in a SARS-CoV-2 RT-PCR-positive 61-year-old male on ICU day 22 with an initial presentation to the ICU that included a $PaO_2$ of 7.6 kPa (normal reference range 8.0–13 kPa), $PaCO_2$ 3.6 kPa (normal reference range 4.6–6.0 kPa), C-reactive protein 242 mg/L (normal reference range <3 mg/L), interleukin-6 154 ng/l (normal reference range <7 ng/L), TNF-α 10.2 ng/L (normal reference range <12 ng/L), AST 1.08 μkat/L (normal reference range <0.76 μkat/L), and D-dimer 0.94 mg/L FEU (age-adjusted cut-off <0.61 mg/L FEU). In this MRI lung perfusion study, the maximum contrast bolus concentration in the pulmonary artery came 15 s after injection, and the maximum contrast bolus concentration reached the aorta 3.4 s later (Fig. 2a, b). Using these values, we calculated a time-to-peak (TTP) map of the lungs, describing how long time the injected

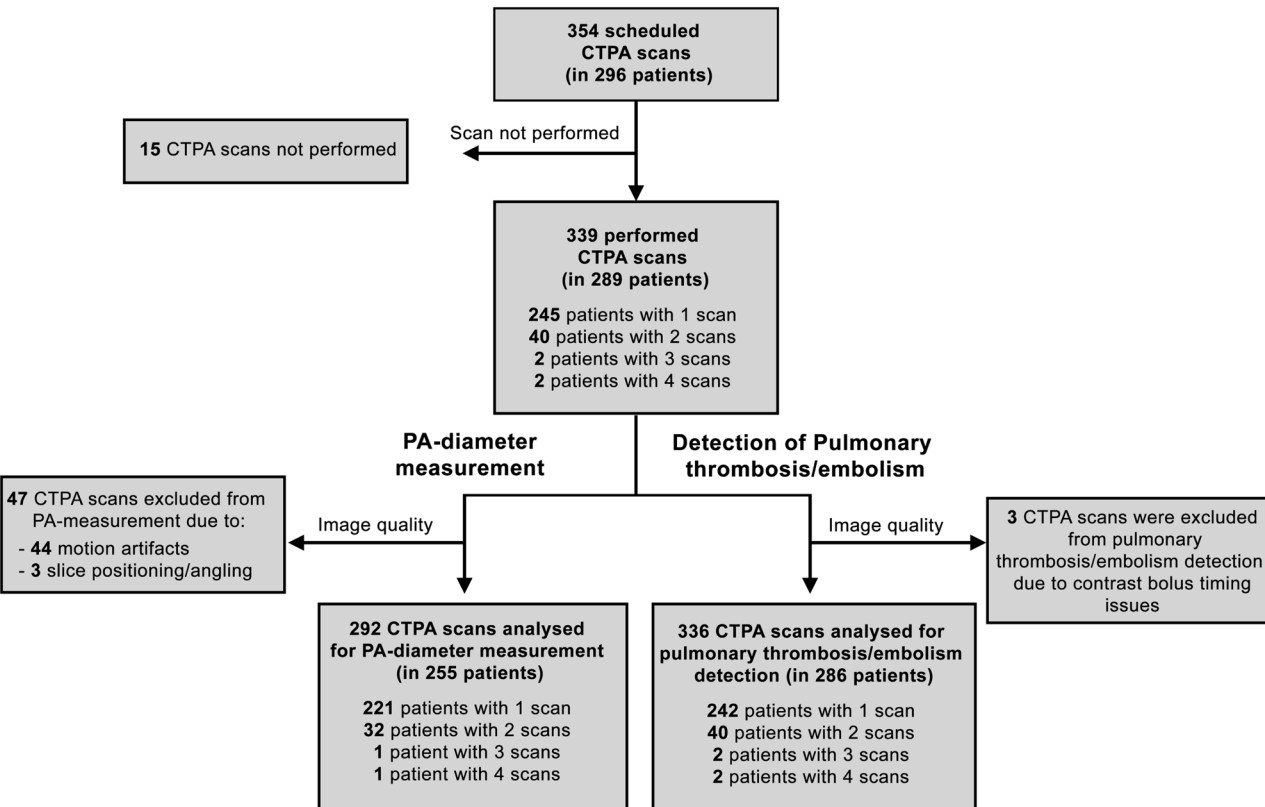

**Fig. 1 Flowchart of the CTPA cohort and subsequent analyses.** CT: computed tomography; CTPA: computed tomography pulmonary angiography; PA: pulmonary artery.

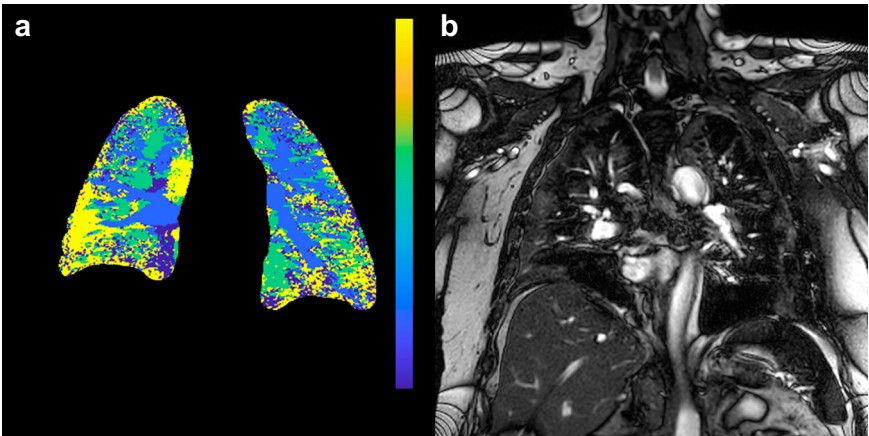

**Fig. 2 MRI of a patient with COVID-19. a** Time-to-peak map of the patient's lung. The color lookup table has been set to maximum blue for pulmonary artery peak and yellow for aortic peak or later. Note large yellow areas dominating the periphery of the lungs, signifying late or no arrival of contrast. **b** Reference coronal T2-weighted scan to identify infiltrates. Reviewed together, perfusion disturbances are apparent both within lung infiltrates and in the normal-appearing pulmonary parenchyma.

contrast took to arrive at a specific area (Fig. 2a). The in vivo perfusion parametric map presented in Fig. 2 demonstrates late contrast arrival, in many areas later than the aortic arrival time. To further understand these paradoxical values, we measured regions of interest in the normal-appearing lung tissue without infiltrates, as identified by $T_2$-weighted anatomical images. The lungs' peripheral regions did not receive a contrast bolus even 10 s after the contrast had peaked in the aorta. We manually segmented the lung and calculated the proportion of the lung with a peak contrast enhancement after the aorta. In 44% of the lungs, there was a delayed or absent contrast peak, suggestive of

microvascular occlusion resulting in the observed impaired pulmonary circulation.

**Summary of clinical results**. We show multimodal evidence of raised pulmonary artery pressure supported by invasive pressure monitoring, frequent pulmonary thromboembolism and a disturbance of blood perfusion on functional MRI. In the cohort of patients undergoing CTPA examination, we found no clear evidence of cytokine levels indicative of cytokine release syndrome, despite a potential indication bias that would skew the cohort characteristics towards those patients most ill. To further explore

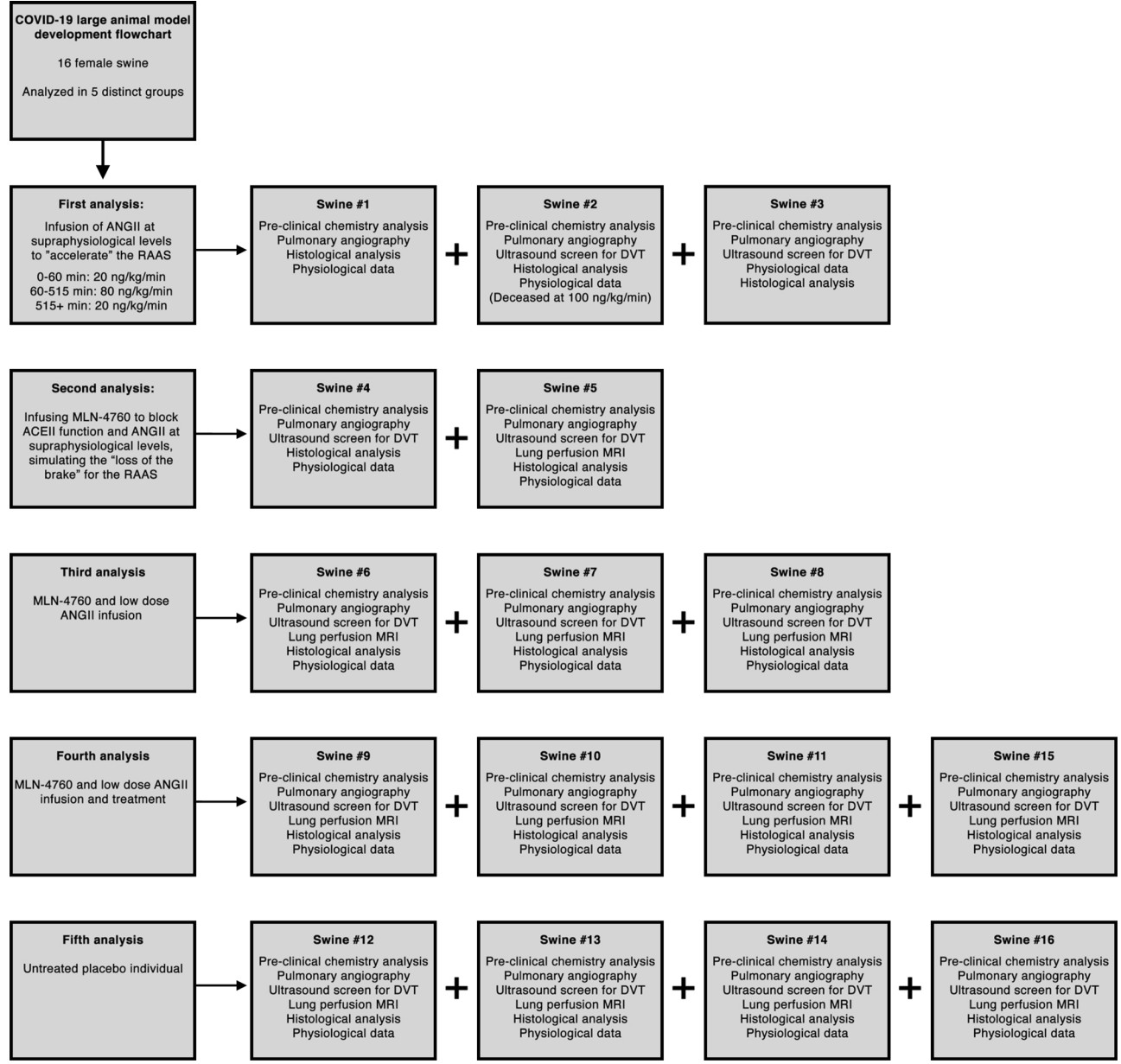

**Fig. 3 Flowchart of preclinical experiments.** ACE2: Angiotensin-converting enzyme 2; ANGII: Angiotensin II; RAAS: Renin-Angiotensin-Aldosterone System; DVT: Deep Venous Thrombosis; MRI: Magnetic Resonance Imaging.

the possible genesis of these findings, we turned to large animal experiments.

**Large animal infusion of supraphysiological levels of angiotensin II.** Based on the above-mentioned findings, it seemed probable that at least part of the COVID-19 pathology involved vasculature disturbances in the lung. We therefore designed a number of experiments, with an overview presented in Fig. 3. Initially, we designed a hypothesis-generating experiment where we infused ANGII in three sedated swine targeting a systolic arterial pressure of 150 mmHg. Within five minutes, arterial and PA blood pressures started to climb (Supplementary Fig. 1). All three swine reached systolic PA pressures exceeding 30 mmHg during the experiment, and one swine died of acute right ventricular heart failure with extensive pulmonary thrombosis/embolism. Meanwhile, $PaO_2$ tended to decrease paralleled by a trend of increased $PaCO_2$ (Supplementary Fig. 2). We started screening for deep venous thrombosis from swine #2 by

ultrasonography to exclude pulmonary embolization, but we could not detect any in this group (Supplementary Table 1), suggestive of in situ thrombus formation in the pulmonary circulation, rather than embolisms. We acquired samples for histology from macroscopically wedge-shaped blood discolored areas. After hematoxylin and eosin staining, we observed severe blood stasis, thickening of alveolar septa, debris in the alveolar sacs and diffuse alveolar damage.

We performed bleeding time assessments in swine #2 and #3 with baseline bleeding times of 285 s and 255 s, respectively. These were reduced to <120 s after 90 min of ANGII infusion and remained low for the remainder of the experiment. The von Willebrand factor activity (GP1bA) increased directly after the initiation of the infusion. In this hypothesis-generating experiment, D-dimer increased, and paradoxically, fibrinogen also increased or remained unchanged despite increased D-dimer. Interleukin-6 and TNF-α remained below the level of detection in all swine during the experiment. Osmolality

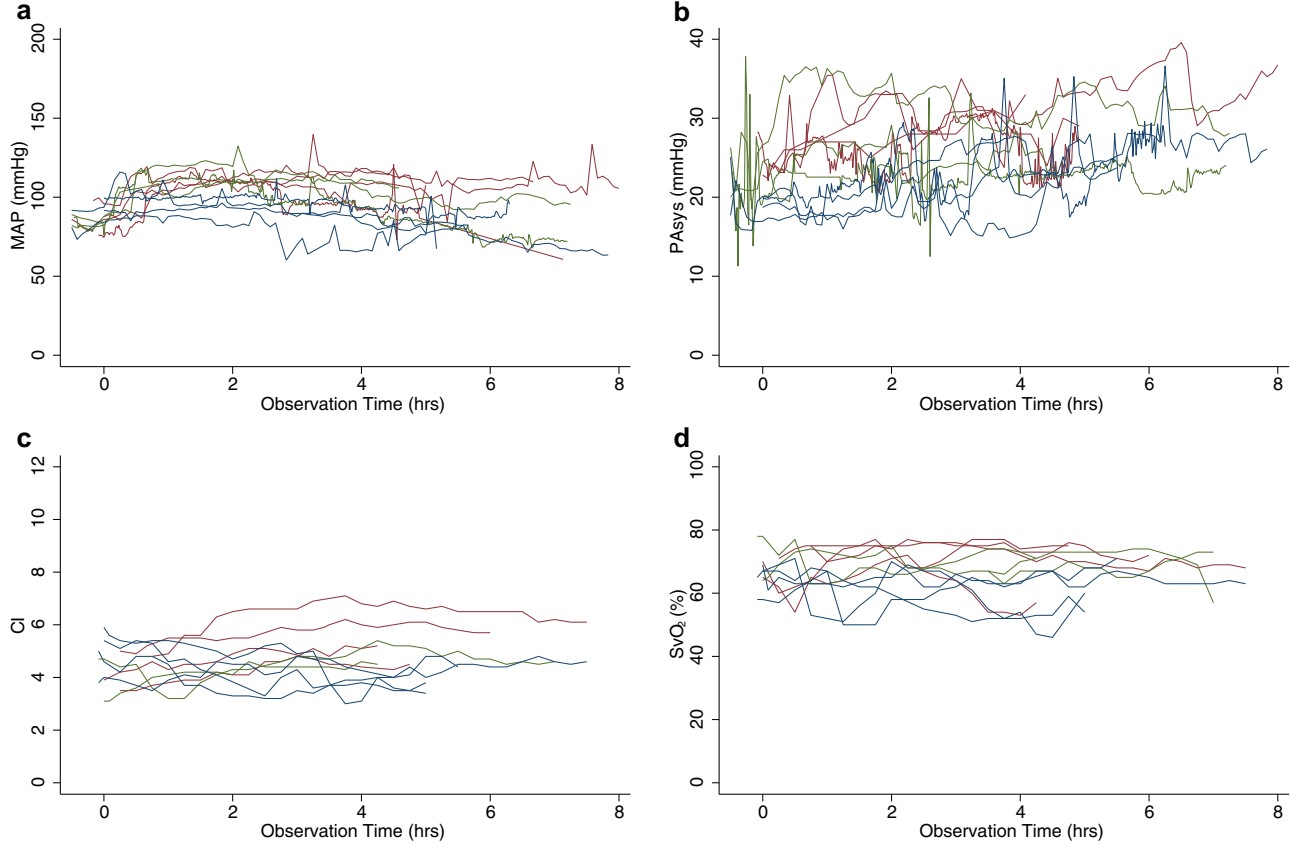

**Fig. 4 Physiological measurements by groups across time. a** Mean arterial pressure. **b** Systolic pulmonary artery pressure. **c** Cardiac index. **d** Mixed venous saturation. In all graphs, red represents individuals in the model group, blue represents individuals in the control group and, green represents individuals in the treatment group. Source data are provided as source data file.

successively increased in all three swine by > 11 mosmol/kg. We present all clinical chemistry data individually in Supplementary Table 1.

Based on these initial experiments, we concluded that infusion of supraphysiological levels of ANGII seemed to produce a remarkably unhealthy state in the swine. Since we hypothesized that COVID-19 patients have a reduction of ACE2 activity, we performed the next set of experiments by administering the ACE2 inhibitor MLN-4760.

**Large animal blocking of ACE2 combined with low-rate infusion of angiotensin II.** We injected the ACE2 inhibitor MLN-4760 and started a low dose infusion of ANGII in four swine. For brevity, we will refer to this group as the model group. We also sedated four additional animals as a control group that received vehicle injections and infusions. We observed a significant increase in systolic pulmonary artery pressure for the model group (Fig. 4a). At three hours into the experiment, systolic pulmonary artery pressure in the model group was significantly higher with an estimated difference of 7.1 (0.95 CI: 0.63–14) mmHg compared to the control group ($n = 11$, $p = 0.03$, linear mixed error-component model with multiple comparisons, model coefficients in Supplementary Table 2) (Fig. 4b). Cardiac index and mixed venous saturation did not differ between the groups (Fig. 4c, d). During the experiment, $PaO_2$ was significantly lower in the model group with an estimated difference of $-2.1$ (0.95 CI $-3.9$–$-0.34$) kPa ($n = 11$, $p = 0.02$, linear mixed error-component model with multiple comparisons, model coefficients in Supplementary Table 2, Fig. 5a). No significant differences in $O_2$ saturation were observed (Fig. 5b). $PaCO_2$ was significantly

higher in the model group with an estimated difference of 0.80 (0.95 CI 0.093–1.5) kPa ($n = 11$, $p = 0.03$, linear mixed error-component model with multiple comparisons, model coefficients in Supplementary Table 2) (Fig. 5c).

To better understand these physiological differences, we performed lung MRI perfusion (Fig. 6). We used the same definition for MRI perfusion disturbance as in the patient, i.e., a contrast bolus peak later than the aorta or no peak at all. We calculated TTP parametric maps of the model group (Fig. 6a), control group (Fig. 6b) and treatment group (Fig. 6c). After approximately five to eight hours (mean time to MRI scan 368 min, $p = 0.94$ for difference between groups, ANOVA), the proportion of the lungs with perfusion disturbance was significantly higher in the model group, $17 \pm 1.2\%$ (Fig. 7a, b), relative to the control group, $10 \pm 4.3\%$. ($n = 11$, 0.95 CI for difference 1.1–13, $p = 0.02$, ANOVA with multiple comparisons). The mean normalized overall contrast peak (reflecting the average timing of lung contrast enhancement) was significantly higher in the model group, $0.73 \pm 0.058$, relative to the control group, $0.54 \pm 0.040$ ($n = 11$, 0.95 CI for difference $-0.29$–$-0.071$, $p = 0.003$, ANOVA with multiple comparisons) (Fig. 7c). The normalized mean range is between 0 (pulmonary artery peak) and 1 (aorta), and physiologically, we would expect it to be somewhat higher than 0.5 due to arterial inflow from the aorta to the pulmonary circulation. In the dynamic contrast series, we also noted that there was stasis and slow outflow of contrast enhancement in dependent areas. The lung fraction with stasis was more extensive in the model group at $13 \pm 3.9\%$ than controls at $6.3 \pm 3.9\%$, albeit not significantly so ($n = 11$, 0.95 CI for difference $-19$–6.6, $p = 0.27$, ANOVA).

**a**

**b**

**c**

Control    Model    Treatment

**Fig. 5 Individual blood gas analysis in arterial blood across time. a** Partial pressure of oxygen. **b** Blood oxygen saturation. **c** Partial pressure of carbon dioxide. Red diamonds represent model individuals, blue control individuals, and green treatment individuals. Source data are provided as source data file.

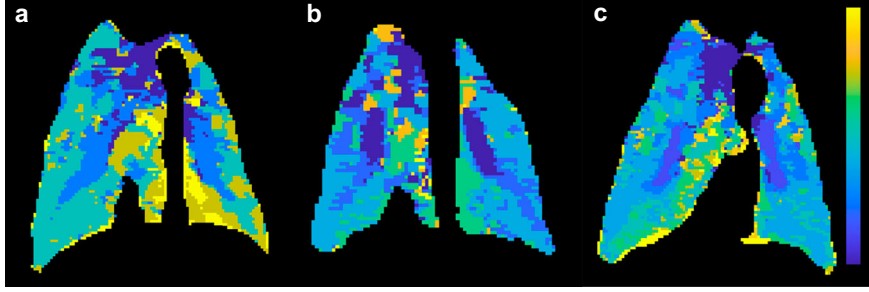

**Fig. 6 Swine MRI perfusion. a** Time-to-peak map of a model swine lung that has received ACE2 inhibition by MLN-4760 and low dose ANGII with the color lookup table set to maximum blue for pulmonary artery peak and yellow for aortic peak or later. **b** Control swine that has been sedated and performed MRI lung perfusion at approximately the same sedation time as the swine in **a**. **c** Treatment swine that has been given oral angiotensin receptor blocker and low molecular weight heparin.

The hourly urine production tended to be lower in the model group ($41 \pm 25$ ml/hour) compared to the control group ($210 \pm 120$ ml/hour), albeit the difference was not significant ($n = 11$, 0.95 CI for difference $-352–11$, $p = 0.07$, ANOVA with multiple comparisons). Four hours into the experiment, the estimated bleeding time was 156 s shorter in the model group as compared to controls, a difference that was statistically significant ($n = 11$, 0.95 CI for difference: $-301–-12$, $p = 0.03$, linear mixed error-component model with multiple comparisons, model coefficients in Supplementary Table 2). Interestingly, we noted that the

surface body temperature was higher in the model group. We could discern no clear pattern in D-dimer or fibrinogen except for one individual who also suffered a groin hematoma during the placement of the arterial introducer (#16) and already had elevated levels at baseline and then coagulation in the test tube.

We performed a macroscopic postmortem evaluation (Fig. 8a–d), followed by histological analyses (Fig. 8e–t). The main histological finding was marked stasis of pulmonary blood vessels, primarily in dependent regions corresponding to our MRI findings (Fig. 8e–h). In the model lungs, we found areas with

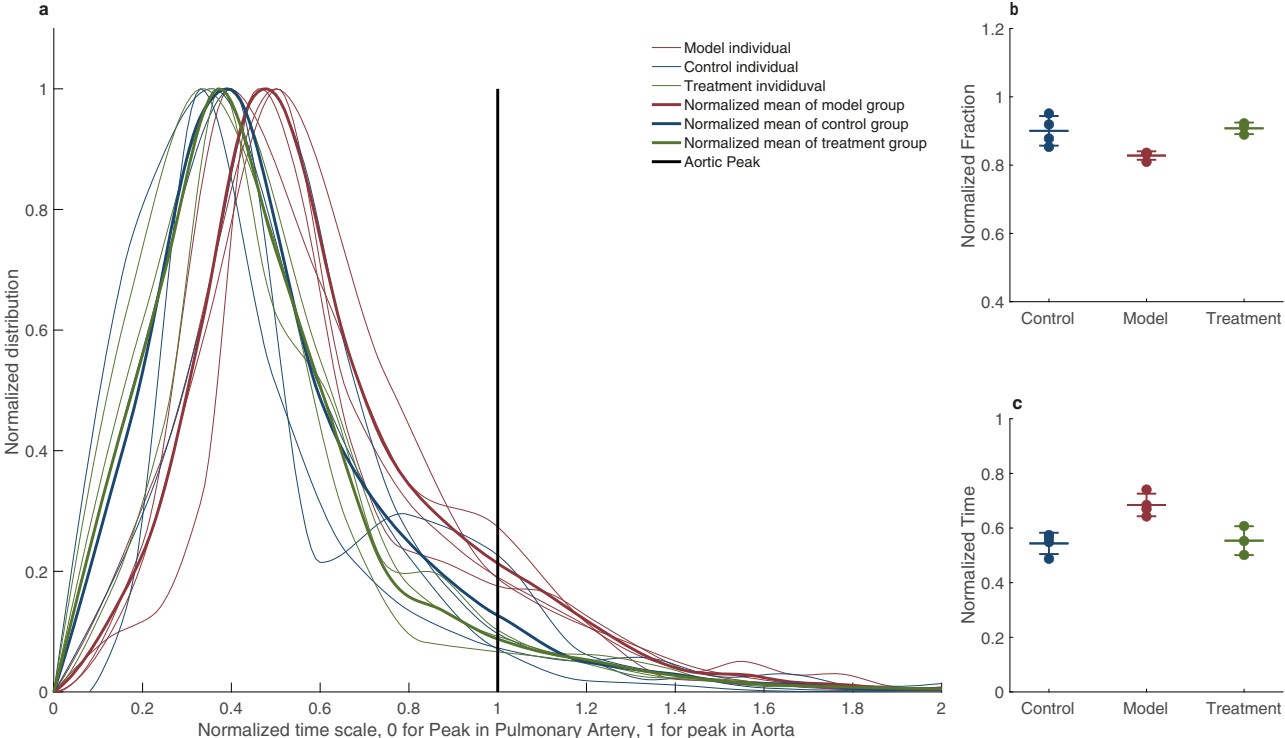

**Fig. 7 Time-to-peak distribution in lungs of swine. a** The distribution of peaks in every individual is plotted with the thinner lines and a solid line represents the mean peak distribution throughout the lungs. In all graphs, red represents the model group, blue represents the control group and, green represents the treatment group. The vertical black bar represents the aorta peak and the cutoff for functional ratio, calculated as the proportion of lung with delayed perfusion (peaking after the aorta). **b** Individual functional ratios represented by dots and a mean bar with standard deviation whiskers for each group of control ($n = 4$), model ($n = 4$), and treatment ($n = 3$). **c** The mean time-to-peak of the lungs with each individual as a scatter plot and mean and standard deviation as whiskers. Source data are provided as source data file.

sloughing of pneumocytes and proteinaceous fluid in the alveolar sacs, consistent with the early signs of diffuse alveolar damage within the areas containing blood vessel stasis. We further found microscopic bleedings in these areas (Fig. 8f). The areas of diffuse alveolar damage corresponded to areas with marked late contrast arrival in time-to-peak parametric maps and perfusion stasis. We also found areas of entirely consolidated lungs, with bronchi filled with fluids and cellular debris (Fig. 8i, j). In the control animals, in similar, smaller areas, we found some fluid in the alveolar sacs and indications of sloughing of pneumocytes but to a lower degree than in the model group (Fig. 8g, h). Even in small sub-millimeter areas of fully consolidated lungs, the bronchi were predominantly not fluid-filled in the control animals (Fig. 8k, l). In the model kidneys, we found minor sloughing of tubular cells consistent with acute tubular necrosis (Fig. 8m, n). We could not identify signs of acute tubular necrosis in the control kidneys (Fig. 8o, p). Further, we could not identify any signs of histological damage in the liver or small bowel (Fig. 8q–t).

To further explore our hypothesis, we also combined the ACE2 inhibitor MLN-4760 with the supraphysiological ANGII infusion regime ($n = 2$). This experiment leads to a severely malignant model with a severe elevation of both systemic arterial and primarily pulmonary arterial pressures that would not be long-term compatible with life (Supplementary Fig. 1). Macroscopically, we found sizable wedge-shaped discolored lung areas in the group receiving supraphysiological ANGII, suggestive of lung infarcts and thus a more malignant disease model.

**Inhibition of RAAS imbalance**. Since introducing RAAS imbalance induced a pathophysiological state in the swine with similarities to that of severe COVID-19, we designed an

experiment to evaluate if this imbalance could be pharmacologically mitigated. We administered 200 mg losartan, as treatment, via an orogastric tube, combined with subcutaneous administration of 10,000 IU of low molecular weight heparin in three swine. For brevity, we will refer to this group as the treatment group.

There were no clear physiological difference between the treatment and model groups during the experiment's initial hours. After approximately 2 h, systemic pressures started to drop towards the control group, and pulmonary pressures never reached as high as the model group (Fig. 4a). The estimated systolic PA pressure decreased significantly faster for the treated group compared to the model group, a difference of $-0.87$ mmHg/h ($n = 11$, 0.95 CI: $-1.54$–$-0.21$, $p = 0.01$, linear mixed error-component model with multiple comparisons, model coefficients in Supplementary Table 2, Fig. 4b). While the model group suffered a decrease in $O_2$ saturation in the arterial blood gases throughout the experiment, the treatment group experienced no decline (Fig. 5a). The estimated differences in $O_2$ saturation and $PaO_2$ were not significant.

We also performed lung MRI perfusion and used the same definition for perfusion disturbance as before (Figs. 6, 7). Relative to the control group, the treated group exhibited a lower fraction of the lungs with perfusion disturbance ($9.2 \pm 1.7\%$ vs. $17 \pm 1.2\%$, $n = 11$, 0.95 CI for difference 15–1.3%, $p = 0.02$, ANOVA with multiple comparisons). However, there was no significant difference between the treatment group and control group ($10 \pm 4.3\%$, $n = 11$, 0.95 CI for difference $-0.074$–0.060, $p = 0.98$, ANOVA with multiple comparisons, Fig. 7b). The mean normalized peak in the treatment group was $0.55 \pm 0.053$ (Fig. 7c). This value is significantly lower than the model group at $0.73 \pm 0.058$ ($n = 11$, 0.95 CI for difference 0.05–0.29, $p = 0.007$,

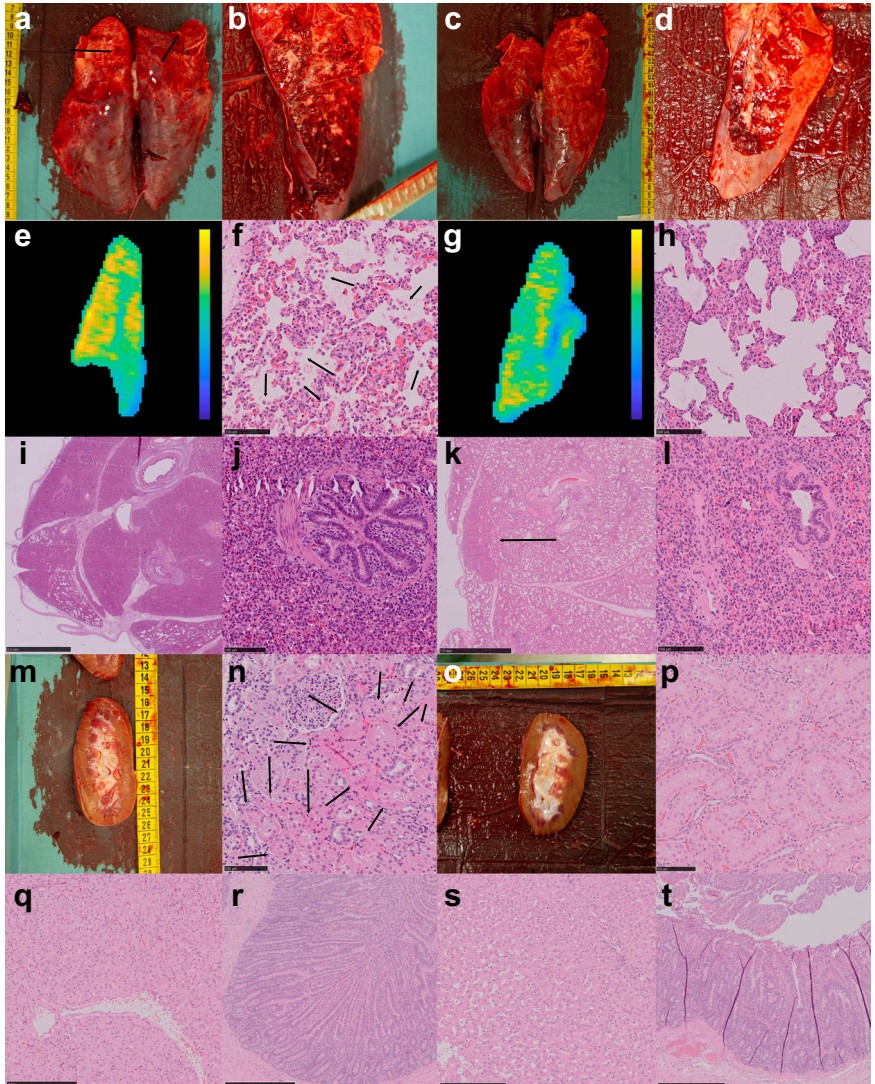

**Fig. 8 Macroscopic and histological postmortem examination. a** Model lungs. Image of posterior aspect with discoloring. **b** Coronal cut through the posterior aspect of the model lung. **c** Control lungs. Image of posterior aspect. **d** Similar coronal cut of a control lung. **e** The dominating microscopic image in the model lungs was blood stasis in vessels of all sizes. This corresponded to areas with stasis perfusion as illustrated by a parametric max enhancement map from the same model animal (yellow is higher contrast enhancement). **f** High power magnification (x40) with fluids in the alveolar sacs and sloughing of pneumocytes into the alveolar space (marked by arrows), consistent with early diffuse alveolar damage (scale bar 100 μm). **g** Parametric relative contrast enhancement, with same color lookup table setting as in **e**, of control animal. **h** High power magnification showing a fluid collection in control lung (marked by arrow) in the alveolar sacs (scale bar 100 μm). **i** The macroscopic texture of the model lung was rubbery. Large areas of the model lungs were fully consolidated (scale bar 2.5 mm). **j** Fluids and cells in bronchi in a fully consolidated model lung (scale bar 100 μm). **k** In small areas in the most posterior aspects of the model lungs we could also identify fully consolidated lung (arrow, scale bar 2.5 mm). **l** Ventilated bronchi in fully consolidated control lungs (scale bar 100 μm). **m** Model kidney. **n** Histological sample from model kidney with narrowing of proximal tubule wall thickness and sloughing of epithelial cells (arrows), consistent with acute tubular necrosis (scale bar 100 μm). **o** Control kidney. **p** Control kidney without signs of acute tubular necrosis (scale bar 100 μm). **q** Model liver without histological damage (scale bar 250 μm). **r** Model small bowel (scale bar 500 μm). **s** Control liver (scale bar 250 μm). **t** Control small bowel (scale bar 500 μm).

ANOVA, multiple comparisons) and close to and not significantly different from the control group at $0.54 \pm 0.042$ ($n = 11$, 0.95 CI for difference $-0.12–0.11$, $p = 0.99$, ANOVA, multiple comparisons). The mean pulmonary artery diameter tended to be larger in the model group; $25 \pm 1.0$ mm compared to $22.1 \pm 0.24$ mm in the treated group. This difference was not significant ($n = 11$, 0.95 CI for difference $-1.9–7.0$, $p = 0.36$, ANOVA with multiple comparisons).

Urine production was higher in the treatment group at $67 \pm 74$ ml/hour than in the model group $41 \pm 25$ ml/hour, albeit not significant ($n = 11$, 0.95 CI for difference $-220–170$, $p = 0.96$, ANOVA with multiple comparisons). D-dimer increased in both

groups, by, on average, 0.027 mg/L FEU per hour in the model group and 0.022 mg/L FEU per hour in the treated group. Notably, and congruent with our hypothesis-generating experiment results, fibrinogen also increased for both groups by 1.07 mg/L per hour in the model group and 0.47 mg/L per hour in the treated group (Supplementary Table 1). We also measured a reduction in bleeding times in both model and treatment groups without significant differences (Supplementary Table 1).

Echocardiography of two swine, one treated and one without treatment, showed significant abnormalities in the latter, suggestive of pulmonary hypertension, with the same functional pattern observed in patients described above. The swine without

treatment showed right ventricular (RV) enlargement, interventricular septal flattening, tricuspid regurgitation peak gradient >30 mmHg, RV free wall hypokinesia, and evidence of RV dysfunction.

**Summary of preclinical results**. We demonstrate that activation of the RAAS both through infusion of ANGII and inhibition of ACE2 leads to a pathophysiological phenotype closely resembling that found in COVID-19. We can induce a reduction in lung perfusion, $O_2$-saturation in the arterial blood gases and PA-pressures, providing mechanistic support for therapeutic strategies involving the RAAS and coagulation. Furthermore, we can ameliorate this pathophysiological state by pharmacological treatments currently under evaluation in clinical trials.

## Discussion

Our results demonstrate similarities between the manifestations of patients with COVID-19 admitted to the ICU and the effects of RAAS overactivation in swine. ANGII has previously been used in the ATHOS III clinical trial[32]. Sub-analyses of this trial highlighted significant thrombotic and infectious complications associated with ANGII[33]. If SARS-CoV disturbs the expression, distribution or function of the ACE2-receptor to produce a subsequent increase in ANGII[2], it could be argued that the same mechanism of receptor disruption from SARS-CoV-2 would lead to some of the effects observed in the animals in this work.

A hallmark of severe COVID-19 is ventilation pathology. We show that this is likely closely associated with coagulopathy and elevations of pulmonary artery pressure. Axial PA diameter measurements are a relatively sensitive marker for detecting even borderline pulmonary hypertension with a sensitivity of 80% and a specificity of 62% when using the cutoff of ≥28 mm[29]. These results are in accordance with similar studies which have found enlarged segmental pulmonary arteries, lung perfusion and microvascular abnormalities in patients with COVID-19[34–40]. On lung perfusion MRI, both in a COVID-19 patient and in our swine model without the virus, there was a marked reduction and, in some areas, likely cessation of blood flow even in the absence of infiltrates. Taken together, our findings provide multimodal evidence of pulmonary blood flow disturbances in COVID-19.

We further show that manipulation of RAAS balance in swine, by infusion of supraphysiological levels of ANGII or by ACE2 blockade and low-level infusion of ANGII, leads to a pathological phenotype that shares several features of COVID-19. In itself, this finding would warrant caution in using ANGII as a vasopressor for COVID-19 patients, something that is now reported[41]. A limitation of this study is that we did not measure RAAS components in our imaging cohort. This was not possible due to the retrospective nature of that part of the study. There are however other reports; a recent study of 82 individuals with COVID-19 showed higher ANGII than in controls and a linear relationship between high ANGII levels and poor clinical outcome[18]. In contrast, renin levels remained similar, suggesting that more specifically the increase in ANGII is closely related to the SARS-CoV-2 infection. Two other studies from another cohort showed no difference in ANGII but significantly lower ANG(1-7) levels[20,21]. In light of those studies, our large animal model blocking ACE2, coupled with a low infusion of ANGII, seems like a plausible model of a possible RAAS dysregulation in COVID-19, which could then be used for testing treatment interventions, as also demonstrated in the present study. The effector mechanisms of SARS-CoV-2 are debated, and it could be argued that a limitation to our large animal studies is that we do not infect the swine with the virus. However, viremia is seldom found in patients, and the panorama of COVID-19 is hard to explain by

an exclusive direct viral infection of cells or cytokine release syndrome[22]. We are suggesting that the pathophysiological syndrome of severe COVID-19 may, therefore, in part, be explained as a downstream effect of a disruption of the RAAS. This work does not provide conclusive evidence about how much RAAS disruption might contribute to COVID-19. We only show that by inhibiting the enzyme responsible for cellular access for SARS-CoV-2, we can induce a pathophysiological syndrome that shares several features with COVID-19, including elevated pulmonary artery pressures, reduction in blood oxygenation, disturbances in the lung circulation, shortened bleeding time, diffuse alveolar damage, and acute tubular necrosis.

In conclusion, we consider the clinical data presented in this work to strengthen the hypothesis that severe COVID-19 is, in part, a vascular syndrome with a hypercoagulable state. If our assumptions are correct, that SARS-CoV-2 introduces a RAAS imbalance, and further, we can simulate this pathophysiological state in swine then this work provides mechanistic support to ongoing clinical trials targeting RAAS imbalance.

## Methods

**Ethical considerations**. The human studies and case reports were approved by the Swedish Ethical Review Authority (no. 2020-01895 and no. 2020-01752); Informed consent was officially waived by the Review Authority due to the retrospective nature of the study, though written informed consent by next of kin was obtained for publication of the one clinical case of MRI lung perfusion and the other for the PA pressure measurements. The human studies and case reports were performed in compliance with the Helsinki declaration. All animal studies were conducted according to Karolinska Institutet guidelines for animal experiments and approved by the Regional Ethics Committee for Animal Research in Stockholm, Sweden (no. 6716-2020).

**Computed tomography pulmonary angiography**. A consecutive cohort of all patients (83 females, 28%; mean age 59 years, standard deviation 16 years) with RT-PCR-confirmed COVID-19 undergoing CTPA at Karolinska University Hospital in Huddinge, Stockholm, Sweden, between March 2nd (first patient admitted) and May 20th were retrospectively evaluated by two raters (T.G., radiologist; J.A., 4th-year medical student) according to a standardized scheme[42], after consultation of a senior thoracic radiologist (M.K.). The ratings were performed independently by the two raters on anonymized data, blinded to the clinical information. The ratings of the radiologist were considered the gold standard and the inter-rater agreement was assessed by intraclass correlation coefficient (two scans were not available for inter-rater analysis due to rejection of image quality by the second rater but not by the gold standard rater). The inter-rater agreement was excellent ($n = 290$, ICC = 0.97, $P < 0.001$). All CTPA imaging was performed on Siemens SOMATOM Definition Flash (Siemens Healthineers, Erlangen, Germany), GE Discovery CT750 HD and Revolution CT (GE Healthcare, Milwaukee, USA) according to clinical routine and scanner allocation was based on the clinical and logistical needs.

**Echocardiography**. Echocardiography data was retrospectively analyzed in a sub-sample ($n = 50$) of patients undergoing CTPA imaging by a senior clinical physiologist (G.A.) to extract the maximal tricuspid regurgitation velocity, estimated systolic pulmonary artery pressure and right ventricular outflow tract acceleration time, as previously described, with normative values from the literature[30,31]. In four patients, only the central venous pressure could be estimated but it is not reported since many patients were intubated. In two patients, none of the parameters were obtainable due to insufficient technical image quality. All echocardiography was performed on GE Vivid S70 (GE Healthcare, Milwaukee, USA).

**Clinical chemistry**. Laboratory data (with reference ranges in parenthesis) on D-dimer (cut-off <0.50 mg/L FEU, with age-adjusted cut-off of patient age x 0.01 mg/L FEU in patients older than 50 years), fibrinogen (2.0-4.2 g/L), platelet count (females 165–387, males 145–348 ×10^9/L), C-reactive protein (<3 mg/L), triglycerides (0.45–2.6 mmol/L), interleukin-6 (<7 ng/L), TNF-α (<12 ng/L), interleukin-10 (<5 ng/L), and serum osmolality (280–300 mosmol/kg) were collected retrospectively on all patients with RT-PCR-confirmed COVID-19 in the CTPA cohort and clinical cases. Lab data were extracted based on temporal proximity (±3 days) to undergoing the CTPA. All samples had been analyzed at Karolinska University Laboratory, accredited according to ISO 15189 by the Swedish Board for Accreditation and Conformity Assessment. Coagulation parameters were analyzed on the Sysmex CS-5100 System (Siemens Healthineers, Erlangen, Germany), platelet counts were analyzed on the Sysmex XN (Siemens Healthineers, Erlangen, Germany), biochemistry analyses were analyzed on Cobas 8000 Analyzer (Roche Diagnostics, Basel, Switzerland), TNF-α and interleukin-10 were analyzed on

Immulite 1000 (Siemens Healthineers, Erlangen, Germany) and osmolality in serum was analyzed with Osmometer Advanced 2020 Multi-Sample (Advanced Instruments, Norwood, USA).

**Patient magnetic resonance perfusion**. A 1.5 Tesla Philips Ingenia MRI scanner, software version R 5.4 (Philips Healthcare, Best, Netherlands) had been used to scan one patient. Anatomical $T_2$-weighted images were used to identify lung infiltrates (technical parameters: MultiVane, 2D field-of-view 460 × 460 mm, voxel size 0.9 × 0.9 × 4 mm, echo/repetition times at 100/2458 ms with a compressed sensing-sensitivity encoding factor 2, 1 average and 36 slices). For the dynamic contrast series, a 4D time-resolved MRI angiography with a keyhole $T_1$-weighted gradient-recalled-echo had been used (field-of-view 500 × 500 mm, voxel size 1.5 × 1.5 × 4 mm, echo/repetition times at 1.8/3.78 ms, a compressed sensing-sensitivity encoding factor 3.6, 1 average with 16 dynamic phases and 60 slices). A Max 3 contrast injector (Ulrich Medical, Ulm, Germany) had been used to administer gadolinium-based contrast agent: gadobutrol solution, 1.0 mmol/ml, 2 ml, followed by 20 ml 0.9% saline solution with 5 ml/s. The first phase acquired the entire K-space in 8.3 s and subsequent phases used a keyhole acceleration with 20% scanning, resulting in a temporal resolution of 1.7 s per phase. TTP maps were generated using the $T_1$ MRI perfusion application in Philips Intellispace v10.1.3 (Philips Healthcare, Best, Netherlands) and for verification, the same analysis was performed using MATLAB (version R2020b, The Mathworks Inc., Natick, USA).

**Clinical and preclinical measurement of PA pressure**. A 7.5 F Swan-Ganz catheter (Edwards Lifesciences, Irvine, USA) for measurements of PA pressure had been placed in the reported patient and in all of the swine. In the patient, it had been placed from the jugular vein and in the swine from the femoral vein using x-ray fluoroscopy and connected to a HemoSphere or Vigilance monitor (Edwards Lifesciences, Irvine, USA).

**Swine experimental setup**. This study took place at the Karolinska Experimental Research and Imaging Center, Karolinska University Hospital, Stockholm, between May 11th and June 23rd, 2020 with additional experiments performed between November 9th and November 19th. Sixteen female swine with weights between 34 and 41 kg were used in this study. Each animal fasted for 12 h with free access to water before the procedure. They arrived sedated after premedication with intra-muscular cepetor vet 1 mg/ml-zoletil 100 (Vetmedic/Virbac, Thirsk, UK) 0.8–1 mg/kg. Induction of anesthesia was conducted with pentobarbital (Sandoz, Holzkirchen, Germany) 1–3 mg/kg and fentanyl (B. Braun, Melsungen, Germany) 2.5 µg/kg as an intravenous bolus dose. Maintenance of anesthesia was achieved with continuous infusion of pentobarbital (0.1–0.2 mg/kg/min) and morphine (Meda, Solna, Sweden) (0.1–0.25 mg/kg/h) titrated to a moderate depth of anesthesia. No muscle relaxants were used during the experiment.

Intubation using an endotracheal tube (7.0) was performed after induction of anesthesia and during spontaneous breathing. The animals were placed in a supine position for ease of access to all catheters and normo-ventilated by pressure-controlled ventilation with Siemens Servoventilator 900 C (Siemens Healthineers, Erlangen, Germany) with a total tidal volume of 10 mL/kg and with an inspiratory oxygen fraction ($FiO_2$) of 0.21[43].

End-tidal $CO_2$ was monitored with capnography incorporated in the surveillance system (Datex, GE Healthcare, Milwaukee, USA). Ventilation and respiratory frequency were adjusted to maintain end-tidal $CO_2$ between 4.7–5.3 kPa. A 5–lead electrocardiogram was recorded continuously as well as arterial pulse oximetry ($SpO_2$) for peripheral oxygen saturation (Datex, GE Healthcare, Milwaukee, USA). The pulse oximeter was placed on the tail. All animals received a urine catheter and hourly diuresis was recorded. Prior to skin incision, local anesthesia (lidocaine 10 mg/ml without adrenaline, Aspen Nordic, Ballerup, Denmark) <0.4 mg/kg) were given. Bleeding time was assessed by cutting the skin of the ear for 10–15 mm and removing any blood every 15 s. When no further bleeding could be identified, this was determined to be the bleeding time. A 7 F central venous catheter (Therumo, Tokyo, Japan) was inserted by open technique via the left internal jugular vein for administration of drugs. For an overview, please see Fig. 3 of the experimental groups. ANGII infusion started at 20 ng/kg/min in swine #1-5 and were incrementally elevated to 80 ng/kg/min in 60 min, followed by a stepwise increase to a maximum dose of 240 ng/kg/min after 515 min except for swine #2 who suffered acute right ventricular heart failure at a dose of 100 ng/kg/min and died. Swine #4 and #5 were administered with the ACE2 inhibitor MLN-4760 with a starting dose of 20 ng/kg/min which was stepwise increased to a maximum 640 ng/kg/min after 150 min; then decreased to 320 ng/kg/min for 15 min; then stopped while a subsequent infusion of ANGII started at 20 ng/kg/min and similarly stepwise increased to 640 ng/kg/min during 275 min. Swine #6-8 and #15 were administered a bolus dose of 1 mg MLN-4760 then subsequently administered infusion of ANGII at a constant dose of 20 ng/kg/min for the rest of the experiment. Swine #9-11 followed the lastly aforementioned protocol but were administered 10.000 U low-weight molecular heparin subcutaneously and 200 mg Losartan through the nasogastric tube 30 min before administering the bolus of MLN-4760. Four normal control animals were also included using the same anesthesia and ventilator strategies.

**Preclinical imaging**. A Philips XD20 angiographic system and 3DRA workstation (Philips Healthcare, Best, Netherlands) was used to place the catheters. For endovascular access, we punctured the vessels using a micropuncture set (Merit Medical AB, Stockholm, Sweden) guided by ultrasonography with Siemens Acuson Sequoia 512 (Siemens Healthineers, Erlangen, Germany). Access was established with an introducer in the femoral artery and the femoral vein. We used a 7 F introducer for the guide and a 8 French introducer for the PA catheter (Terumo, Tokyo, Japan). We placed the 7.5 F PA catheter (Edwards Lifesciences, Irvine, USA) by fluoroscopy guidance and the location was confirmed by invasive pressure measurement. A distal access guide (Envoy 6 F, Cordis, Santa Clara, USA) was then placed in the proximal part of the descending aorta for continuous monitoring of aortic pressure and a 7 F pigtail catheter was placed in the right atrium of swine #2 and #3 in the supraphysiological ANGII infusion group for pulmonary angiography. Ultrasonography was performed every second hour during the experiment to exclude deep venous thrombosis in the hind legs using Siemens Acuson Sequoia 512 (Siemens Healthineers, Erlangen, Germany). Echocardiography was performed on a GE Vivid S70 (GE Healthcare, Milwaukee, USA).

**Preclinical magnetic resonance perfusion imaging**. MRI in the large animal model was performed on a Siemens MAGNETOM Aera 1.5 Tesla scanner (Siemens Healthineers, Erlangen, Germany) using a clinical body coil. An anatomical 2D motion-corrected (BLADE) $T_2$-weighted sequence was used to identify lung infiltrates with the following parameters: field-of-view 400 × 400 mm, voxel size 2.1 × 2.1 × 2.0 mm, echo/repetition time at 112/6870 ms, 1 average and 90 slices. For the dynamic contrast series, we used 4D time-resolved MRI angiography (TWIST). The initial two swine undergoing MRI (swine 6 and 7, alternative parameters in parentheses) were scanned with a higher temporal resolution protocol, while the remaining swine were imaged using higher spatial resolution: field-of-view 400 × 400 mm, voxel size 2.1 × 2.1 × 3.0 mm (3.1 × 3.1 × 5.0 mm), 48 slices (30 slices), echo/repetition times at 0.69/1.60 ms (0.57/1.54 ms), flip angle of 20°, GRAPPA acceleration factor of 4, 1 average with 60 dynamic phases, with a resulting temporal resolution of 0.96 s (0.61 s) per phase. Gadolinium was injected by hand: gadobutrol solution, 1.0 mmol/ml, 2 ml, followed by 10 ml 0.9% saline solution with approximately 5 ml/s.

**Pre-clinical chemistry analysis**. Blood samples were acquired at baseline and at 120, 240, 360, 435 min or directly prior to death when systolic blood pressure dropped below 70 mmHg. Arterial blood gases were obtained at baseline, 75, 270, 345, 435, 545, 665, 750 min or when close to circulatory collapse. EDTA whole-blood, citrated platelet-poor plasma and serum samples were analyzed using proprietary assays at the Karolinska University Laboratory, accredited according to ISO 15189 by the Swedish Board for Accreditation and Conformity Assessment. Coagulation parameters were analyzed on the Sysmex CS-5100 System (Siemens Healthineers, Erlangen, Germany), hematological analyses were analyzed on the Sysmex XN (Siemens Healthineers, Erlangen, Germany), chemistry analyses were analyzed on the Cobas 8000 Analyzer (Roche Diagnostics, Basel, Switzerland), TNF-α and interleukin-10 were analyzed on Immulite 1000 (Siemens Healthineers, Erlangen, Germany) and osmolality in serum was analyzed with the Osmometer Advanced 2020 Multi-Sample (Advanced Instruments, Norwood, USA).

**Pre-clinical histological analysis**. Lungs samples were placed in 4% formaldehyde at +8 °C for 24 h. The formaldehyde was then exchanged. Samples were embedded, cut and stained using Mayer's Hematoxylin and Eosin 0.2% (Histolab AB, Stockholm, Sweden). Representative slides were scanned using a NanoZoomer S60, (Hamamatsu Photonics, Japan). Images were then evaluated using NDP.view 2 for MacOS (Hamamatsu Photonics, Japan).

**Statistics and reproducibility**. All values reported are mean and standard deviations if not explicitly stated to be median and interquartile range. Statistical analyses were performed using MATLAB (version R2020b, The Mathworks Inc., Natick, USA), IBM SPSS Statistics version 25 for Mac (IBM, Armonk, USA) and Stata 16.1 (StataCorp 2019, College Station, Texas, USA). A threshold of $P < 0.05$ was considered statistically significant. Statistical analysis of manually drawn regions of interest in normal-appearing lung tissue on MRI perfusion were analyzed using MATLAB and imtool3D developed by Justin Solomon imtool3D (https://www.mathworks.com/matlabcentral/fileexchange/40753-imtool3d), MATLAB Central File Exchange, retrieved April 25, 2018) with further in-house development for 4D and ROI measurements. Extrapulmonary and extra-mediastinal tissues were manually masked. ROIs were manually selected in the superior portion of the pulmonary artery and in the aorta/innominate artery bifurcation. TTP and max enhancement parametric maps were generated by using the MATLAB function max and the color lookup table used for the figures is the parula color scheme from MATLAB. Mean relative contrast enhancement was defined as the enhancement between 10 to 25 s after the aortic peak divided by baseline signal during the initial 10 s. Stasis was estimated as the fraction with relative contrast enhancement above noise defined as three standard deviation of the signal during the measurement volumes. Other descriptive statistics on MRI perfusion were also generated in MATLAB. To generate Fig. 7, curves were interpolated using the modified Akima method. This was done to generate a

unified *x*-axis for simultaneous plotting. No interpolation was performed prior to calculating the functional ratio, mean TTP values or max relative enhancement of the lungs. Inter-rater agreement of CTPA measurements were calculated using the intraclass correlation coefficient of average measurements in SPSS.

Two-way ANOVA with Dunn-Sidak correction for multiple comparisons where used since there is an uneven number of individuals to compare differences between multiple groups. Every individual is only sampled once prior to ANOVA. Longitudinal data were fitted using Stata's 'mixed' command with restricted maximum likelihood and an unstructured covariance matrix. Equations were designed with the response variable of a physiological parameter or clinical chemistry data with fixed effects investigated for group status and random intercepts and slopes for each individual. The formula can be written as follows. Let $y_{it}$ denote the response variable for animal j (j = 1, 2, 3, … 6) and let $time_{ij}$ be the time at which measurement i was taken from animal j. Then the mixed model can be written as

$$y_{ij} = \beta_0 + \beta_1*time_{ij} + \beta_2*group_j + \beta_3*time_{ij}*group_j + e_{ij} \qquad (1)$$

where:

$\beta_0 = \beta_0 + u_{0j}$

$\beta_1 = \beta_1 + u_{1j}$

$\beta_0$ is the overall intercept, $\beta_1$ the overall slope; $u_{0j}$ represent the individual animal-specific random intercept, $u_{1j}$ the individual animal specific random slope, group is a categorical variable representing group membership (reference category: control group). Group is interacted with linear time to capture differences in growth rates between groups. Slopes were estimated only for those measurements for which time resolution was high. $e_{ij}$ is the time-specific residual capturing unmeasured time-varying characteristics on the individual level. *p*-values were corrected for multiple comparisons using the Sidak-Holm adjustment.

A minimum of four histological samples from lungs of all individuals were taken and sectioned. Samples were taken from the lower lobe at the dependent area, and roughly equidistant along the lateral wall to the lobe border and a final sample from the superior lobe. At least two samples were collected from the superior pole of the right kidney, inferior border of liver and small bowel with the most distension or discoloration that could be located. Representative macroscopic images including images of cut sections were acquired from three swine.

**Reporting summary**. Further information on research design is available in the Nature Research Reporting Summary linked to this article.

## Data availability
The data that support the findings of this study are available from the corresponding author upon reasonable request except for the patient imaging raw data due to privacy concerns.

Source data are provided with this paper.

## Code availability
MRI image analysis has been performed in MATLAB using the superclass perfusion and subclass T1 perfusion available at github.com/SWICUrays/MRItools (https://doi.org/10.5281/zenodo.4492927). The code used for analysis has commit hash 63c46c4. To view, designate inflow and outflow ROIs, a modified version of Imtool3D is provided. To view 4D-volumes and parametric maps, the class awObj is recommended.

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

## Acknowledgements
We would like to thank Pellina Jansson, Johanna Doshé, Britt Meijer, Ann-Christine Sandberg-Nordqvist, Anette Ebberyd, Niklas Bark, Lisbeth Söderblom and Laszlo Szekely for excellent logistic and technical assistance. Furthermore, we would like to acknowledge the helpful assistance regarding MRI by Magnus Tengvar and Thelma Gudmundsdóttir at the Department of Radiology Solna, Karolinska University Hospital. We would also like to acknowledge colleagues from the Department of Neuroradiology, Karolinska University Hospital, for critical discussion regarding perfusion imaging. Finally, we would like to thank Stina Englesson for assistance with graphical illustrations. TG was supported by grants from the Stockholm Region ALF and Clinical Postdoc programs. JL was supported by grants from the Stockholm Region Clinical Postdoc program, a private donation by Tedde Jeansson Sr and MedTechLabs at Karolinska Institutet.

## Author contributions
S.R. and J.L. initiated the study. J.A., M.K., R.O., T.G. and J.L. analyzed computed tomography pulmonary angiography data. G.A. analyzed both clinical and preclinical echocardiography data. S.R. and F.C.J. performed the clinical pulmonary artery catheter measurements and analyzed the data. S.R. and F.C.J. stressed the need for adequate imaging in COVID-19 and took care of the ICU patient during the clinical MRI. R.V.P. and J.L. optimized the clinical MRI protocol. S.N. and J.L. interpreted the clinical MRI study. S.R., M.J.F. and J.L. designed the animal studies and together with J.A. performed the animal studies and analyzed data. R.O., M.P., T.G. and J.L. optimized the preclinical MRI protocol. R.O., M.P. and T.G. performed the preclinical MRI scans; T.G. and J.L. analyzed the data. J.A., R.O., T.G. and J.L. performed pre-clinical histological evaluation. A.S. identified the need for interleukin analyses and analyzed the data. S.R., A.S., M.F., M.J.F. and J.L. analyzed clinical chemistry data. H.E., R.O., T.G. and J.L. performed statistical analysis. T.G., M.J.F. and J.L. supervised the study. J.L. wrote the first draft and all authors revised the manuscript critically.

## Funding

## Competing interests
The authors declare no competing interests.
