## [Peer Review File · Nature Communications]

REVIEWER COMMENTS

Reviewer #3 (Remarks to the Author):

Thanks to Rysz and colleagues for providing this revised manuscript. This is a combined clinical observational study and a series of large animal experiments. The central thesis of the authors is that RAAS imbalance plays a role in the pathophysiology of severe COVID-19. In the resubmission the authors have expanded the animal experiments to include several treatment groups. This is a well written manuscript, which has been enhanced by additional details provided in the methods section. While each component part is of interest, the authors fail to establish a causal link between the clinical observations and those made in the animal model. The manuscript requires the reader to accept that ANG-II mediates the pulmonary vascular dysfunction evident in the clinical observations at face value.

In addition, the retrospective observational nature of the clinical data is open to substantial confounding. This, coupled with the absence of relevant control groups in the animal experiments, undermines any conclusions which may be drawn. While derangement of RAAS in COVID-19 may be a contributory factor to its pathophysiology, these data do not prove the hypothesis.

Reviewer #4 (Remarks to the Author):

I have read with interest the paper by Lundberg et al. which gives new evidence on the role of the RAS system in the development of Covid 19 through a translational approach. The retrieved insights seem to support the concept that an imbalance in RAS might play a direct role in the development of pulmonary complication of Covid 19 independently of primary inflammatory and prothrombotic mechanisms. The limited number (12) of animals used without control groups) and the characteristics patients enrolled (which are not described satisfactorily in the text), however, might represent a limitation to the conclusions and affect the priority of this paper as well as it does the lack of a causal proven inference. Therefore, the paper is to be considered more hypothesis generating rather than conclusive.

With regards to specific comments raised by reviewer 1:

The authors have partially amended the first comment, since they still do not provide images of normal lungs. Despite this, the explanations given in the text are detailed enough. On the contrary, the authors have been asked to tabulate data about enrolled subjects but I could not find this kind of table in the text, which would be useful to summarize clinical characteristics.

Comments three and four have been amended.

Regarding comment 5-6: the paper is now easy to follow and well balanced. The lack of data on RAS peptide levels in patients enrolled, however, stands as a major limit which still needs to be acknowledged in the discussion.

Additional comment: Legend to Figure 2 should be amended since it seems to refer to four columns, whereas there are just three columns.

Reviewer #5 (Remarks to the Author):

Comments for the Authors:

Overall, the authors are to be congratulated for a substantial amount of work, and for the degree of novelty presented. However, the ms approach is not particularly systematic, and the number of observations (pre-clinical) is rather limited in terms of presenting a degree of reproducibility of the findings.

- What are the noteworthy results?

To an extent, this study presents novel preliminary evidence for some of the mechanism

underlying COVID-19, currently assumed in some clinical trials.

- Will the work be of significance to the field and related fields? How does it compare to the established literature? If the work is not original, please provide relevant references. Some of the work is original, i. e. the part on the intervention. In my opinion, one of the weaknesses of the study is in the rather limited systematic approach to the data collection. While the timeliness of generating these results is helpful, their interpretation is as strong as the power of the data, which are based on interesting observations, but on a rather limited number of animals.

- Does the work support the conclusions and claims, or is additional evidence needed? While common sense may suggest that the evidence presented could support the conclusions and claims, I think that a greater number of observations could increase the confidence in the results presented.

- Are there any flaws in the data analysis, interpretation and conclusions? - Do these prohibit publication or require revision? I think that the data analysis is appropriate, pending some considerations in terms of the statistical power associated with the limited number of animals studied. This power determines the degree of support for the interpretation of the results and the overall conclusions.

- Is the methodology sound? Does the work meet the expected standards in your field? The authors are to be congratulated for the extensive work performed, which could be completed by a relatively small group of teams globally. Having said that, the ms in its current form is rather difficult to follow due to the variety of approaches taken, and the way in which the text is structured.

- Is there enough detail provided in the methods for the work to be reproduced? Mostly.

Please find additional and more detailed comments in the attached pdf file.

Overall, there is some novelty that could be of interest to the wider public, yet the current form of the ms is rather difficult to digest, due in part to poor organisation, and in part to the limited systematic approach, which may affect reproducibility of the findings.

I hope that the above comments may help to improve the ms.

Yours sincerely,
Federico Formenti

King's College London
University of Oxford
Wadham College, Oxford

We would like to first begin by thanking the Editors for their considerations and the many Reviewers for their valuable comments that have helped us to further strengthen the manuscript. Please find our point-by-point responses below. We have additionally attached the manuscript with and without tracked changes.

Reviewer #3 (Remarks to the Author):

Reviewer: Thanks to Rysz and colleagues for providing this revised manuscript. This is a combined clinical observational study and a series of large animal experiments. The central thesis of the authors is that RAAS imbalance plays a role in the pathophysiology of severe COVID-19. In the resubmission the authors have expanded the animal experiments to include several treatment groups.

This is a well written manuscript, which has been enhanced by additional details provided in the methods section. While each component part is of interest, the authors fail to establish a causal link between the clinical observations and those made in the animal model. The manuscript requires the reader to accept that ANG-II mediates the pulmonary vascular dysfunction evident in the clinical observations at face value.

Authors: We again thank Reviewer #3 for valuable comments on our revised manuscript. We agree that our experiments do not unequivocally prove a causal link between the clinical observations and those made in the animal models. However, the primary objective of the animal experiments was to enhance our understanding of whether a RAAS imbalance could be a contributor to COVID-19 pathophysiology. In May, when we began these animal experiments, the main discussion regarding ACEi and ARB was whether such treatment could be harmful in COVID-19 patients. The then reigning dogma seemed to lean towards discontinuation of such therapies during COVID-19 care. Clinical studies now support the continued use of RAAS blockade, if permitted by blood pressure.

Before initiating the study, based on the literature, our hypothesis was that a RAAS imbalance in COVID-19 could mediate some of the adverse pathophysiological effects. However, we did not expect the deleterious effects observed in our first hypothesis-generating experiment with supraphysiological ANGII infusion. In this revised manuscript, we have clarified that we now show that if we, inhibit the ACE2 enzyme and introduce a RAAS imbalance, we induce a pathophysiological state in the swine that shares several features with COVID-19 rather than a causal link. We have additionally updated the Introduction and Discussion to reflect some of the current debate and pivotal findings regarding the use of ACEi and ARB in COVID-19 accordingly.

In addition, the retrospective observational nature of the clinical data is open to substantial confounding. This, coupled with the absence of relevant control groups in the animal experiments, undermines any conclusions which may be drawn. While derangement of RAAS in COVID-19 may be a contributory factor to its pathophysiology, these data do not prove the hypothesis.

Authors: We agree that acquiring additional data would facilitate the interpretation of the findings. We have therefore to the very best of our abilities during the on-going pandemic and with consideration of the ethical principles of research in animal models obtained additional data, which corroborates our initial results. We have now added three control

animals sedated in the same way, with the same ventilator and anesthesia strategy. We show that MLN-4760 inhibition of ACE2 produces a physiological state with significantly higher pulmonary artery pressure, lower PaO₂, higher PaCO₂, a smaller fraction of late perfused lung, higher mean time to peak, and shorter bleeding times compared to treated and control animals.

Reviewer #4 (Remarks to the Author):

I have read with interest the paper by Lundberg et al. which gives new evidence on the role of the RAS system in the development of Covid 19 through a translational approach. The retrieved insights seem to support the concept that an imbalance in RAS might play a direct role in the development of pulmonary complication of Covid 19 independently of primary inflammatory and prothrombotic mechanisms. The limited number (12) of animals used without control groups) and the characteristics patients enrolled (which are not described satisfactorily in the text), however, might represent a limitation to the conclusions and affect the priority of this paper as well as it does the lack of a causal proven inference. Therefore, the paper is to be considered more hypothesis generating rather than conclusive.

Authors: We agree with this sentiment, echoed by Reviewer #3, and by doing so we have now appended a control group (n=4) sedated with the same anesthesia and ventilator strategy. We have collected physiological data and MRI imaging of these control animals and updated the manuscript accordingly. We have also incorporated adjustments for multiple comparisons in our statistical analyses. Consequently, we have reduced some of the statistical power by introducing multiple comparisons. We have further added an animal in the MLN-4760 and low dose ANGII group to facilitate a more structured postmortem analysis with a thorough documentation of macroscopic pathology completed by an experienced professor in clinical pathology who has experience in COVID-19. We agree with both original Reviewer #3 and Reviewer #4 that we should have included this control group directly in the initial revision. However at the time, our primary clinical motivation was to evaluate if oral ARB could potentially block the effect and facilitate clinical therapeutic intervention in those suffering from COVID-19, if our initial hypothesis were to be right. Scientifically, this was an oversight, one that has now been remedied per your recommendations.

With regards to specific comments raised by reviewer 1:

The authors have partially amended the first comment, since they still do not provide images of normal lungs. Despite this, the explanations given in the text are detailed enough. On the contrary, the authors have been asked to tabulate data about enrolled subjects but I could not find this kind of table in the text, which would be useful to summarize clinical characteristics.

Authors: Macroscopic and microscopic images of lung tissue in the model and controls are now provided in Figure 8 for a more clear comparison of the pathology present in the disease models. We now also have added the mean age (59 ± 16 years) and gender distribution (72% male) to the results.

Comments three and four have been amended.

Regarding comment 5-6: the paper is now easy to follow and well balanced. The lack of data on RAS peptide levels in patients enrolled, however, stands as a major limit which still needs to be acknowledged in the discussion.

Authors: We agree with the Reviewer, that measurements of such peptides would be interesting and strengthen our hypothesis. Due to the retrospective nature of the study and the lack of such testing in the first wave of the pandemic at our institution, we have not yet been able to obtain such data. However, since the submission of this study in July, there have been additional publications providing such additional data reporting RAAS imbalance in COVID-19 patients, which are now referenced and discussed.

Additional comment: Legend to Figure 2 should be amended since it seems to refer to four columns, whereas there are just three columns.

Authors: While we do not understand which part of the legend to Figure 2 is referred to as a fourth column in the previous submission, we have now replaced Figure 2 with a more systematic presentation of the macroscopic and microscopic findings, including comparison of macroscopic images compared to MRI parametric maps and histological analysis of lungs and kidney.

Reviewer #5 (Remarks to the Author):

Comments for the Authors:

Overall, the authors are to be congratulated for a substantial amount of work, and for the degree of novelty presented. However, the ms approach is not particularly systematic, and the number of observations (pre-clinical) is rather limited in terms of presenting a degree of reproducibility of the findings.

- What are the noteworthy results?

To an extent, this study presents novel preliminary evidence for some of the mechanism underlying COVID-19, currently assumed in some clinical trials.

- Will the work be of significance to the field and related fields? How does it compare to the established literature? If the work is not original, please provide relevant references.

Some of the work is original, i. e. the part on the intervention. In my opinion, one of the weaknesses of the study is in the rather limited systematic approach to the data collection. While the timeliness of generating these results is helpful, their interpretation is as strong as the power of the data, which are based on interesting observations, but on a rather limited number of animals.

- Does the work support the conclusions and claims, or is additional evidence needed?

While common sense may suggest that the evidence presented could support the conclusions and claims, I think that a greater number of observations could increase the confidence in the results presented.

- Are there any flaws in the data analysis, interpretation and conclusions? - Do these prohibit publication or require revision?

I think that the data analysis is appropriate, pending some considerations in terms of the statistical power associated with the limited number of animals studied. This power determines the degree of support for the interpretation of the results and the overall conclusions.

Authors: First, we would like to thank Reviewer #5 for their interest in our findings. Since the first four comments are centered around the number of animals, we would like to reply to the comments simultaneously. When performing animal experiments in Sweden, we are mandated by law to acknowledge the 3 Rs, replacement, reduction and refinement. We have used large animals to obtain as much data as possible from the same individual.

Our ethical permit application argued that replacement with small rodents was not possible due to the amount of data collected from the same individual in swine is more comparable to humans, and the pandemic situation mandated rapid research. Further, to perform these comprehensive analyses, we would have required a rather large number of small rodents. Moreover, some of the acquired measurements are not possible in small rodents, e.g., invasive pulmonary artery monitoring with a Swan-Ganz catheter. The ethical review board accepted and approved our permit. On the other hand, that gives a much larger weight to the Reduction part of the 3R principle.

We performed the experiments sequentially, as illustrated in the previously submitted Figure S2. Specifically, we consider animals #1-#3 to be hypothesis-generating in nature, whereby they were the only animals included in the first submission. We then proceeded with further hypothesis-generating experiments in animals #4 and #5. As hypothesis-generating experiments, we did not perform statistical analysis on them. We then performed the MLN-4760 + low dose ANGII experiment in swine #6-8. After completing the first experiment with ARB and LMWH (#9), we calculated the power needed for statistical significance, given the tertiary analysis variability. Power analysis is also mandated by our ethical framework and we (correctly) calculated that three large animals in each group would be sufficient to show a significant difference with an alpha level of .05 in MRI lung perfusion, O₂ saturation, and pulmonary artery pressure.

After a dialogue with the Editorial office, we have added a control group of suitable size, with consideration of statistical power and ethical standards. Original Reviewer #3 initially requested an ARDS control group (e.g., LPS, oleic acid, live bacteria, influenza, or a combination). Since original Reviewer #1 asked for normal control animals and Reviewer #4 and #5 also wanted normal control data, we have now provided such a group (n=4). We have also performed an additional experiment with an individual in the MLN-4760 and low dose ANGII group to obtain macroscopic images of the organs systematically acquired by a clinical pathologist experienced with COVID-19.

*- Is the methodology sound? Does the work meet the expected standards in your field?
The authors are to be congratulated for the extensive work performed, which could be completed by a relatively small group of teams globally. Having said that, the ms in its current form is rather difficult to follow do to the variety of approaches taken, and the way in which the text is structured.*

Authors: We thank Reviewer #5 for this positive comment. We have now updated the text and added a number of Figures to increase the readability of the study.

*- Is there enough detail provided in the methods for the work to be reproduced?
Mostly.*

Please find additional and more detailed comments in the attached pdf file.

Authors: We have amended parts of the Methods in accordance with the comments provided in the pdf. We also would like to especially thank Reviewer #5 for the Hannon *et al.* (1989) reference. We were unaware of that paper, and it will most definitely be useful for us in other swine studies.

Furthermore, we have re-written the manuscript according to the recommendations provided in the pdf file, which will benefit the reader. The revision includes the addition of a paragraph in the Introduction regarding the current state of RAAS acting interventions in COVID-19. In the Results section, we have also added comments concerning urinary output and an analysis of bleeding time.

Overall, there is some novelty that could be of interest to the wider public, yet the current form of the ms is rather difficult to digest, due in part to poor organisation, and in part to the limited systematic approach, which may affect reproducibility of the findings.

Authors: We have revised the manuscript for clarity. The introduction of the control group should make the line of thought more succinct.

I hope that the above comments may help to improve the ms.

Authors: They certainly have and we are grateful for them.

*Yours sincerely,
Federico Formenti*

*King's College London
University of Oxford
Wadham College, Oxford*

REVIEWERS' COMMENTS

Reviewer #4 (Remarks to the Author):

The authors have satisfactorily addressed my comments.

The addition of a control group, albeit limited in size, the inclusion of demographic information on the patients and the inclusion of images of lung pathology improved the paper.

Discussion is also improved.

Massimo Volpe

Dept. of Clinical and Molecular Medicine

Sapienza University of Rome, Italy

Reviewer #5 (Remarks to the Author):

Since the initial submission, the authors have generated control data that were certainly necessary, and have also improved the manuscript in terms of presentation.

The authors have satisfactorily addressed my concerns, and the ms discusses the study limitations appropriately now. As far as I can see, the authors have also satisfactorily responded to the other reviewers' concerns.

I only have a couple of minor, formatting suggestions on the supplementary material; I apologise these suggestions were missed in my previous set of comments.

I think that the figures S1 (Physiological measurements for all swine) and S2 (Blood gas analysis in all swine) axes font needs to be increased for readability, and the label and unit of measure for the horizontal axes of figure S2 needs to be presented.

For the data presented that were not recorded continuously (e. g. blood PO₂?), presenting continuous lines linking the measured values may be inappropriate, and it may be preferable to present average and SD results of the values actually recorded (scatter plot, i. e. without linking the average value "artificially", with no evidence that the values in between measurements actually laid on the straight line between them). For example, PO₂ was measured at time 0 hours and 2 or even 4 hours; I suggest presenting these time points, and remove the lines that connect them, as PO₂ could have increased or decreased during the 2- or 4-hour period, in the absence of evidence that the PO₂ change was linear between the two time points.

I hope that the above minor comments may help finalise the ms.

Yours sincerely,

Federico Formenti

King's College London

University of Oxford

Wadham College, Oxford

We want to begin by again thanking the Editors for their considerations and the many Reviewers for their valuable comments that have further strengthened the manuscript. Please find our point-by-point responses below.

Reviewer #4 (Remarks to the Author):

Reviewer: The authors have satisfactorily addressed my comments. The addition of a control group, albeit limited in size, the inclusion of demographic information on the patients and the inclusion of images of lung pathology improved the paper. Discussion is also improved.

*Massimo Volpe
Dept. of Clinical and Molecular Medicine
Sapienza University of Rome, Italy*

Authors: We thank reviewer #4 and agree with the comments about controls animals in particular.

Reviewer #5 (Remarks to the Author):

Reviewer: Since the initial submission, the authors have generated control data that were certainly necessary, and have also improved the manuscript in terms of presentation.

The authors have satisfactorily addressed my concerns, and the ms discusses the study limitations appropriately now. As far as I can see, the authors have also satisfactorily responded to the other reviewers' concerns.

I only have a couple of minor, formatting suggestions on the supplementary material; I apologise these suggestions were missed in my previous set of comments.

I think that the figures S1 (Physiological measurements for all swine) and S2 (Blood gas analysis in all swine) axes font needs to be increased for readability, and the label and unit of measure for the horizontal axes of figure S2 needs to be presented.

Authors: We have increased the font size and rotated image S1 to increase the readability further. We have also added labels and units for the horizontal axis in figure S2.

Reviewer: For the data presented that were not recorded continuously (e. g. blood PO₂?), presenting continuous lines linking the measured values may be inappropriate, and it may be preferable to present average and SD results of the values actually recorded (scatter plot, i. e. without linking the average value "artificially", with no evidence that the values in between measurements actually laid on the straight line between them). For example, PO₂ was measured at time 0 hours and 2 or even 4 hours; I suggest presenting these time points, and remove the lines that connect them, as PO₂ could have increased or decreased during the 2- or 4-hour period, in the absence of evidence that the PO₂ change was linear between the two time points.

Authors: We considered three different types of figures for the blood gas measurement: Scatter plots including each individual separately with no linking of the measurement time points, a line plot with continuous linking of the measurement of each separate individual, and a plot showing the mean and SD of each group. For scatter plots without linking, it is harder to follow each individual's development throughout the experiment and make group comparisons. Please see the figure from the main text without linking.

Also for reference, the supplementary blood gas figure.

For line plots with linking, it is easier to follow each individual's time trend, but as the Reviewer points out, it may create an idea of an "artificial" time trend in-between measurements. Maybe a note at the end of the figure reading, "For readability, individual measurements have been connected with lines", would help. For plots of mean and SD of each group, we hesitate for two reasons. First, it would distort the presentation of the individual time trends. Mean, and SD is especially problematic as in our linear mixed error component model, we estimate the time trends over the experiment for each individual rather than tests of group differences in each point in time. Second, our measurements, especially those around the end of the experiment, were taken at different points in time. Calculating an average for these measurements would distort the minute precision we have in the exact measurement time. We would like to leave the decision of the figures with the editorial office. If the scatter plots are considered preferable, we will immediately provide production quality images and update the Supplementary material pdf.

Reviewer: I hope that the above minor comments may help finalise the ms.

*Yours sincerely,
Federico Formenti*

*King's College London
University of Oxford
Wadham College, Oxford*

Authors: We again thank Reviewer #5 for the helpful comments.